# Cofilin-induced unidirectional cooperative conformational changes in actin filaments revealed by high-speed atomic force microscopy

Kien Xuan Ngo[1†], Noriyuki Kodera[2,3]*[†], Eisaku Katayama[4], Toshio Ando[2,5], Taro QP Uyeda[1,6,7]*

[1]Biomedical Research Institute, National Institute of Advanced Industrial Science and Technology, Tsukuba, Japan; [2]Department of Physics and Bio-AFM Frontier Research Center, Kanazawa University, Kanazawa, Japan; [3]Precursory Research for Embryonic Science and Technology, Japan Science and Technology Agency, Kawaguchi, Japan; [4]Department of Biology, Graduate School of Science, Osaka City University, Osaka, Japan; [5]Core Research for Evolutionary Science and Technology, Japan Science and Technology Agency, Kawaguchi, Japan; [6]Graduate School of Life and Environmental Sciences, University of Tsukuba, Tsukuba, Japan; [7]Graduate School of Medical Life Science, Yokohama City University, Yokohama, Japan

*For correspondence: nkodera@staff.kanazawa-u.ac.jp (NK); t-uyeda@aist.go.jp (TQU)

[†]These authors contributed equally to this work

Competing interests: The authors declare that no competing interests exist.

**Abstract** High-speed atomic force microscopy was employed to observe structural changes in actin filaments induced by cofilin binding. Consistent with previous electron and fluorescence microscopic studies, cofilin formed clusters along actin filaments, where the filaments were 2-nm thicker and the helical pitch was ~25% shorter, compared to control filaments. Interestingly, the shortened helical pitch was propagated to the neighboring bare zone on the pointed-end side of the cluster, while the pitch on the barbed-end side was similar to the control. Thus, cofilin clusters induce distinctively asymmetric conformational changes in filaments. Consistent with the idea that cofilin favors actin structures with a shorter helical pitch, cofilin clusters grew unidirectionally toward the pointed-end of the filament. Severing was often observed near the boundaries between bare zones and clusters, but not necessarily at the boundaries.

## Introduction

Actin filaments are involved in a variety of important functions in eukaryotic cells, including muscle contraction, amoeboid movement, cytokinesis, intracellular transport, and transcriptional regulation within the nucleus. These diverse functions depend on the interaction between actin and specific actin binding proteins (ABPs), and it is generally assumed that specific biochemical signaling is involved in the spatial and temporal regulation of each actin–ABP interaction. During migration of amoeboid cells, for instance, cofilin plays essential roles in continuous extension of lamellipodia by severing actin filaments to promote filament depolymerization or to initiate polymerization from new barbed ends (reviewed by *Bravo-Cordero et al., 2013*). Three independent biochemical mechanisms are known to inhibit cofilin activity; phosphorylation of Ser3, sequestration to phosphatidylinositol 4,5-bisphosphate (PIP$_2$) in plasma membrane, and lower pH (*Bernstein and Bamburg, 2010*; *Bravo-Cordero et al., 2013*). These inhibitory mechanisms are implicated in the regulation of cofilin activity during cell migration, since experimental unleashing of inactive cofilin has been shown to initiate cofilin-dependent processes (*Bernstein and Bamburg, 2010*; *Bravo-Cordero et al., 2013*). Critically speaking, however,

**eLife digest** Actin is one of the most abundant proteins found inside all eukaryotic cells including plant, animal, and fungal cells. This protein is involved in a wide range of biological processes that are essential for an organism's survival. Actin proteins form long filaments that help cells to maintain their shape and also provide the force required for cells to divide and/or move.

Actin filaments are helical in shape and are made up of many actin subunits joined together. Actin filaments are changeable structures that continuously grow and shrink as new actin subunits are added to or removed from the ends of the filaments. One end of an actin filament grows faster than the other; the fast-growing end is known as the barbed-end, while the slow-growing end is referred to as the pointed-end.

Over 100 other proteins are known to bind to and work with actin to regulate its roles in cells and how it forms into filaments. Cofilin is one such protein that binds to and forms clusters on actin filaments and it can also sever actin filaments. Severing an actin filament can encourage the filament to disassemble, or it can help produce new barbed ends that can then grow into new filaments. Previous work had suggested that cofilin severs actin filaments at the junction between regions on the filament that are coated with cofilin and those that are not. It was also known that cofilin binding to a filament causes the filament to change shape, and that the shape change is propagated to neighboring sections of the filaments not coated with cofilin. However, the details of where cofilin binds and how changes in shape are propagated along an actin filament were not known. Furthermore, the findings of these previous studies were largely based on examining still images of actin filaments, which are unlike the constantly changing filaments of living cells.

Ngo, Kodera et al. have now analyzed what happens when cofilin binds to and forms clusters along actin filaments using a recently developed imaging technique called high-speed atomic force microscopy. This technique can be used to directly visualize individual proteins in action. Consistent with previous findings, Ngo, Kodera et al. observed that filaments coated with cofilin are thicker than those filaments without cofilin; and that cofilin binding also substantially reduces the helical twist of the filament. Ngo, Kodera et al. also found that these changes in shape are propagated along the filament but in only one direction—towards the pointed-end. Moreover, cofilin clusters also only grew towards the pointed-end of the actin filament—and the filaments were often severed near, but not exactly at, the junctions between cofilin-coated and uncoated regions. Such one-directional changes in shape of the actin filaments presumably help to regulate how other actin binding proteins can interact with the filament and consequently regulate the roles of the filaments themselves.

these data do demonstrate that active cofilin is required, but there is little experimental evidence that localized activation of cofilin is necessary for proper cofilin functions in cell migration. Indeed, overexpression of constitutively active S3A cofilin did not inhibit motility (*Endo et al., 2003*; *Popow-Wozniak et al., 2012*). Thus, biochemical signaling is not sufficient to explain how cofilin activities are properly regulated spatially and temporally in cells.

On the other hand, many ABPs alter the atomic structure of actin subunits within filaments, and in certain cases these conformational changes are cooperative. For instance, the pioneering work of Oosawa and his colleagues demonstrated that the increase in fluorescence intensity of a fluorophore on actin saturates when only one molecule of the myosin motor domain is added for each 10 actin subunits within filaments (*Oosawa et al., 1973*). Similar myosin-induced cooperative conformational changes have been detected in various other assays (*Tawada, 1969*; *Fujime and Ishiwata, 1971*; *Loscalzo et al., 1975*; *Miki et al., 1982*; *Prochniewicz et al., 2010*). In addition, dense binding of cofilin shortens the helical pitch of actin filaments by 25% (*McGough et al., 1997*; *Galkin et al., 2001*; *Sharma et al., 2011*), and time-resolved phosphorescence anisotropy (*Prochniewicz et al., 2005*) and differential scanning calorimetry (*Dedova et al., 2004*; *Bobkov et al., 2006*) showed that one molecule of bound cofilin changes the structure of ~100 actin subunits within a filament. Moreover, binding of cofilin to actin filaments is cooperative (*Hawkins et al., 1993*; *Hayden et al., 1993*; *McGough et al., 1997*; *De La Cruz, 2005*; *Hayakawa et al., 2014*). This implies that cooperative conformational changes induced by an ABP are propagated to neighboring actin subunits, increasing their affinity for

that, or another, ABP. This could provide a novel mechanism by which actin filaments change their function by regulating their affinities for various ABPs (*Tokuraku et al., 2009*; *Michelot and Drubin, 2011*; *Schoenenberger et al., 2011*; *Uyeda et al., 2011*; *Galkin et al., 2012*; *Romet-Lemonne and Jegou, 2013*).

However, the currently available information on structural changes to actin filaments is limited to high-resolution static images (electron microscopy), low-resolution dynamic changes (fluorescence microscopy), and bulk biochemical analyses. There is little information available in the molecular mechanism that mediates the propagation of structural changes to neighboring actin subunits. Atomic force microscope (AFM) is unique in that it enables detailed structural analysis of wet protein samples (*Müller and Dufrêne, 2008*), and recent dramatic improvements in scanning speed now enables real time imaging of conformational changes in protein samples with high-spatial resolution (*Ando et al., 2013*). This high-speed AFM (HS-AFM) has been used to visualize molecular movements such as the stepping motion of myosin V along actin filaments (*Kodera et al., 2010*), rotary catalysis of $F_1$ ATPase without a rotor (*Uchihashi et al., 2011*), and light-induced conformational changes in bacteriorhodopsin (*Shibata et al., 2010*). Here, we used HS-AFM to directly visualize conformational changes in actin filaments induced by cofilin binding. We found that conformational changes within cofilin clusters unidirectionally propagate to the neighboring bare actin zone in a cooperative manner and that the growth of the cofilin cluster follows this unidirectional cooperative conformational change.

## Results

### Observation of actin filaments on a supported lipid bilayer

For AFM, actin filaments must be immobilized on the stage, yet they must have the freedom of movement to bind cofilin and exhibit the resultant changes in the helical twist that accompany cofilin binding. We therefore formed a bilayer of positively charged lipid on the surface of freshly peeled mica (*Yamamoto et al., 2010*) fixed to the observation stage. A solution of actin filaments was then placed on the supported lipid bilayer and HS-AFM was performed. Right-handed double helical filaments were clearly visualized (*Figure 1*), as in earlier reports (*Weisenhorn et al., 1990*; *Schmitz et al., 2010*).

We approximated the positions of the crossover points, where the two strands of actin filament are aligned vertically, using the position of the highest point (peak) in each half helix identified in the AFM images. Quantitative analysis indicated that the height of those peaks, or the thickness of the filaments, was 8.6 ± 0.8 nm (average ± SD), and the spacing between the peaks, or half helical pitch, was 36.8 ± 4.3 nm (*Figure 1D,E*). These values, which are consistent with previously reported structural data (*Hanson and Lowy, 1963*; *Hanson, 1973*; *Egelman et al., 1982*), will hereafter be referred to as normal height and normal half helical pitch, respectively. The distribution of half helical pitches ranged from ~26 nm to ~45 nm and was well fit by a normal distribution with a standard deviation of 4.3 nm. This distribution should reflect natural variation in half helical pitch, as was proposed on the basis of a similar distribution of half helical pitches in electron micrographs of negatively stained specimens (*Hanson, 1967*; *Egelman et al., 1982*) and also measurement errors. To assess the overall magnitude of the measurement errors, we prepared paracrystals of actin filaments in the presence of 30 mM $MgCl_2$ (*Figure 1C*) (Hanson, 1973). The distribution of the half helical pitches of the paracrystals measured under the same conditions was narrower than that of free filaments, with a mean of 36.5 nm and standard deviation of 3.0 nm (*Figure 1F*). In a hypothetical case where the paracrystals do not undergo spontaneous conformational changes, this distribution is the upper limit of the measurement errors in our system. Furthermore, assuming normal distributions of the true half helical pitches and measurement errors, the true standard deviation of the half helical pitches of control filaments was calculated to be 3.1 nm or larger, by subtracting the variance of the actin paracrystals from that of control filaments. This indicated that the structure of the actin filaments does indeed vary with changes in helical twist.

### Observation of actin filaments fully bound with cofilin

When 75 nM cofilin was added to actin filaments immobilized on the lipid surface, cofilin gradually bound to the actin filaments, and sections of filaments bound with cofilin molecules were easily identified, as the bound filaments appeared thicker and the peaks were taller (*Figure 1B*). Peak heights within long cofilin clusters (longer than eight consecutive half helices) showed a single distribution of 10.6 nm ± 1.0 nm, approximately 2 nm taller than the control filaments (*Figure 1D*). This is consistent with electron microscopic analysis (*Galkin et al., 2011*), which showed that filaments fully decorated

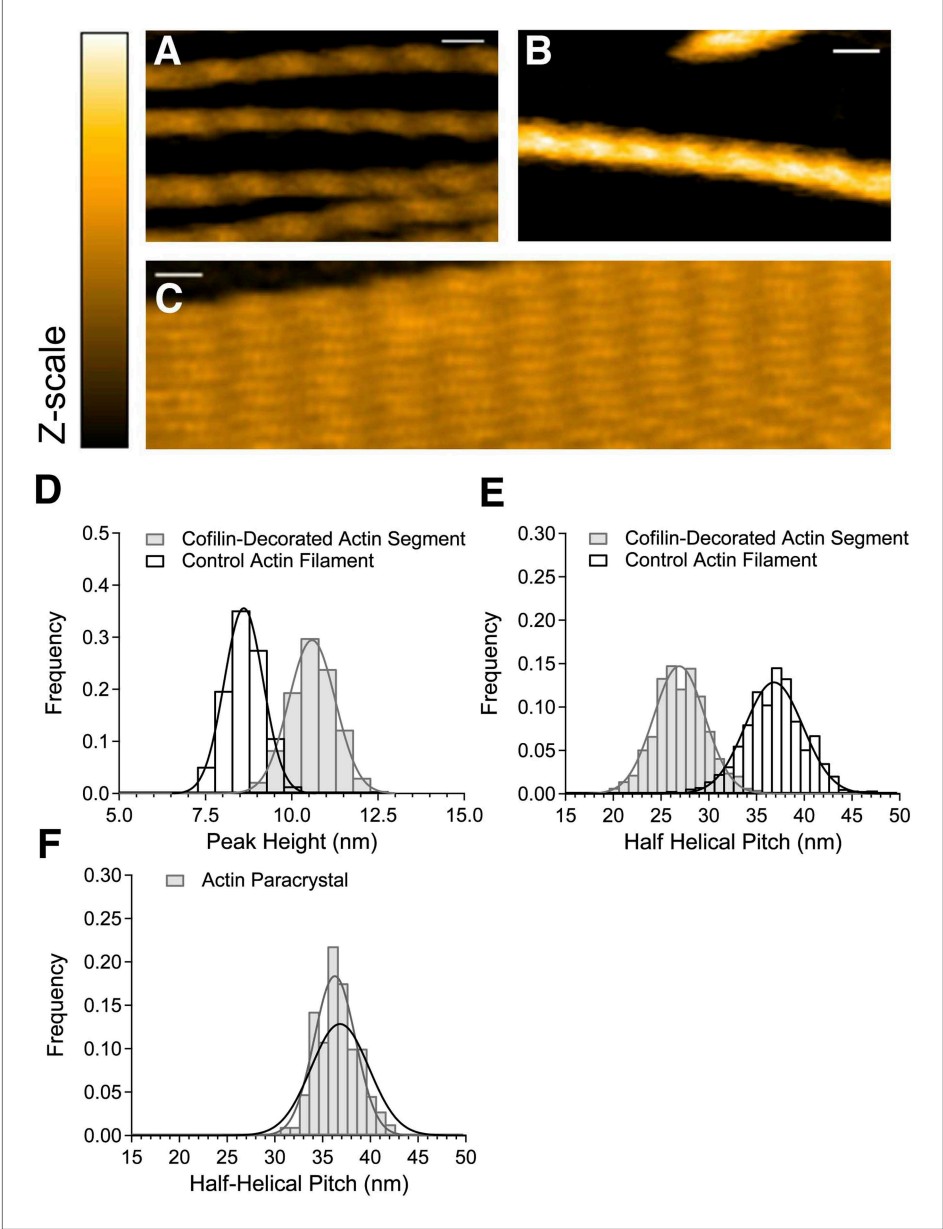

**Figure 1**. HS-AFM observation of control and cofilin-bound actin filaments. (**A**) Control actin filaments without cofilin. (**B**) Actin filaments fully bound with cofilin over an extended distance. (**C**) Paracrystals of actin filaments. Bars: 25 nm, Z-scale: 0–12 nm. (**D** and **E**) Histograms of peak heights (**D**) and lengths of half helical pitches (**E**) in control actin filaments and cofilin-decorated actin segments. N was between 1722 and 2536. (**F**) Half helical pitches of actin paracrystals (N = 1009). Solid lines show Gaussian fittings with confidence intervals of 99.73%. For comparison, the dark line in (**F**) shows the Gaussian fitting of control actin filaments. Measurements were made in F buffer containing 1 mM ATP (**A**), 1 mM ATP, and 75 nM cofilin (**B**) or 1 mM ATP and 30 mM $MgCl_2$ (**C**). Student's $t$-test comparing control and cofilin-decorated actin segments showed that the differences in peak heights and half helical pitches are statistically significant at $p \leq 0.00001$. The mean of the half helical pitches of control actin filaments and paracrystals did not differ significantly. Models of control actin filaments and cofilin-decorated actin filaments with two different orientations on substrates are shown in *Figure 1—figure supplement 1*.

The following figure supplement is available for figure 1:

**Figure supplement 1**. Models of control actin filaments and cofilin-decorated filaments on a flat substrate.

with cofilin are ~2.3-nm thicker than control filaments due to the presence of cofilin molecules (*Figure 1—figure supplement 1*). Moreover, the height distribution was fit well by a single normal distribution, which suggests that within those long cofilin clusters, both actin strands are homogeneously bound with cofilin molecules.

Cofilin binding shortened the half helical pitch by 27% to 26.9 nm (*Figure 1E*), which is again consistent with earlier electron microscopic (*McGough et al., 1997*; *Galkin et al., 2001*) and AFM analyses (*Sharma et al., 2011*). The standard deviation of the distribution of half helical pitches of the fully cofilin-decorated filaments was 3.8 nm, and subtracting the maximum possible measurement error yielded a standard deviation of 2.3 nm or larger. This value is smaller than that of the control filaments (3.1 nm) and suggests that cofilin binding restricts torsional movements of the filaments within cofilin clusters.

## Half helical pitch on either side of the cofilin cluster

We next analyzed shorter cofilin clusters along otherwise apparently bare filaments (*Figure 2*) so that both ends of the cluster were visible within the imaged area. Intriguingly, we noticed that the half helical pitch in the apparently bare section on one side of the cofilin cluster was short, while the pitch on the other side of the cluster was nearly normal.

The observation buffer was then changed to F buffer containing 0.1 mM ATP, 1 mM ADP, hexokinase, glucose, 20 nM subfragment-1 (S1) of skeletal myosin II, and 75–100 nM cofilin, so that the polarity of the actin filaments could be identified from the characteristic tilted binding angle seen when S1 transiently binds the filament (*Huxley, 1963*). This analysis revealed that the half helical pitch of the bare zone on the pointed-end side of the cofilin cluster was short, while that on the barbed-end side was slightly longer than the normal pitch (*Figure 2* and *Table 1*). Student's *t*-test indicated that the differences in the mean half helical pitches between the first neighbor bare zone on either side of the cluster and the control filaments are statistically significant (p < 0.00001 and p < 0.001 for pointed-end and barbed-end side, respectively). Pitches of the neighboring half helices second from the cofilin clusters were nearly normal (*Table 1*).

## Half helical pitch around sparsely bound cofilin

The results summarized above demonstrate that cofilin clusters induce distinctively asymmetric conformational changes in bare zones immediately neighboring the clusters, but it is still uncertain whether a stretch of many bound cofilin molecules, as in clusters, is necessary to induce such conformational changes. To answer that question, we needed to visualize individual cofilin molecules bound to actin filaments, and measure the half helical pitch of the filament around those bound molecules. This has not been possible because cofilin molecules are too small to image individually using electron microscopy or AFM. We therefore engineered a fusion protein in which the N-terminal half of the rod domain of α-actinin was attached to the C-terminus of cofilin. In electron micrographs of negatively stained specimens (*Figure 3A*), as well as in HS-AFM images (*Video 1* and *Figure 3—figure supplement 4*), we were able to see clusters of cofilin-rod with a half helical pitch ~25% shorter than the bare zones, which is similar to control cofilin molecules bound to actin filaments (*Figure 3C*).

Rod-like structures were often observed to stick out from apparently bare sections of the filaments (*Figure 3B*). These are most likely the rod portions of the cofilin-rod molecules bound sparsely along seemingly bare sections of the filaments, because their length was ~10 nm, which is expected for half an α-actinin rod (*Yan et al., 1993*; *Anson et al., 1996*), and because similar rod-like structures were rarely observed in control images of cofilin added to actin filaments (*Figure 3C*). We therefore estimated the helical pitches on both sides of single rod-like structures sticking out from the filaments by measuring the filament length that encompassed three actin subunits along one strand on either side of the bound molecule. In addition, because we did not know the polarity of each filament, we compared the helical pitches on both sides by dividing the longer helical pitch by the shorter one. This yielded a value of 1.02 ± 0.02 (average ± SD, n = 9), which was much smaller than the 1.30 ± 0.18 (n = 4) measured for cofilin clusters in electron micrographs or 1.37 calculated from the AFM data (*Figure 1E*) and was comparable to the value of randomly selected regions along control actin filaments (1.04 ± 0.05, n = 18). Furthermore, the helical pitches around the apparently singly bound cofilin-rod molecules (36.0 ± 1.0 nm, n = 5) did not significantly differ from the control helical pitch (36.3 ± 1.0 nm, n = 8) measured in electron micrographs.

In HS-AFM observations, we were able to observe four cases of an apparently single cofilin-rod molecule binding transiently to an actin filament (*Figure 3D* and *Video 2*). Consistent with the electron

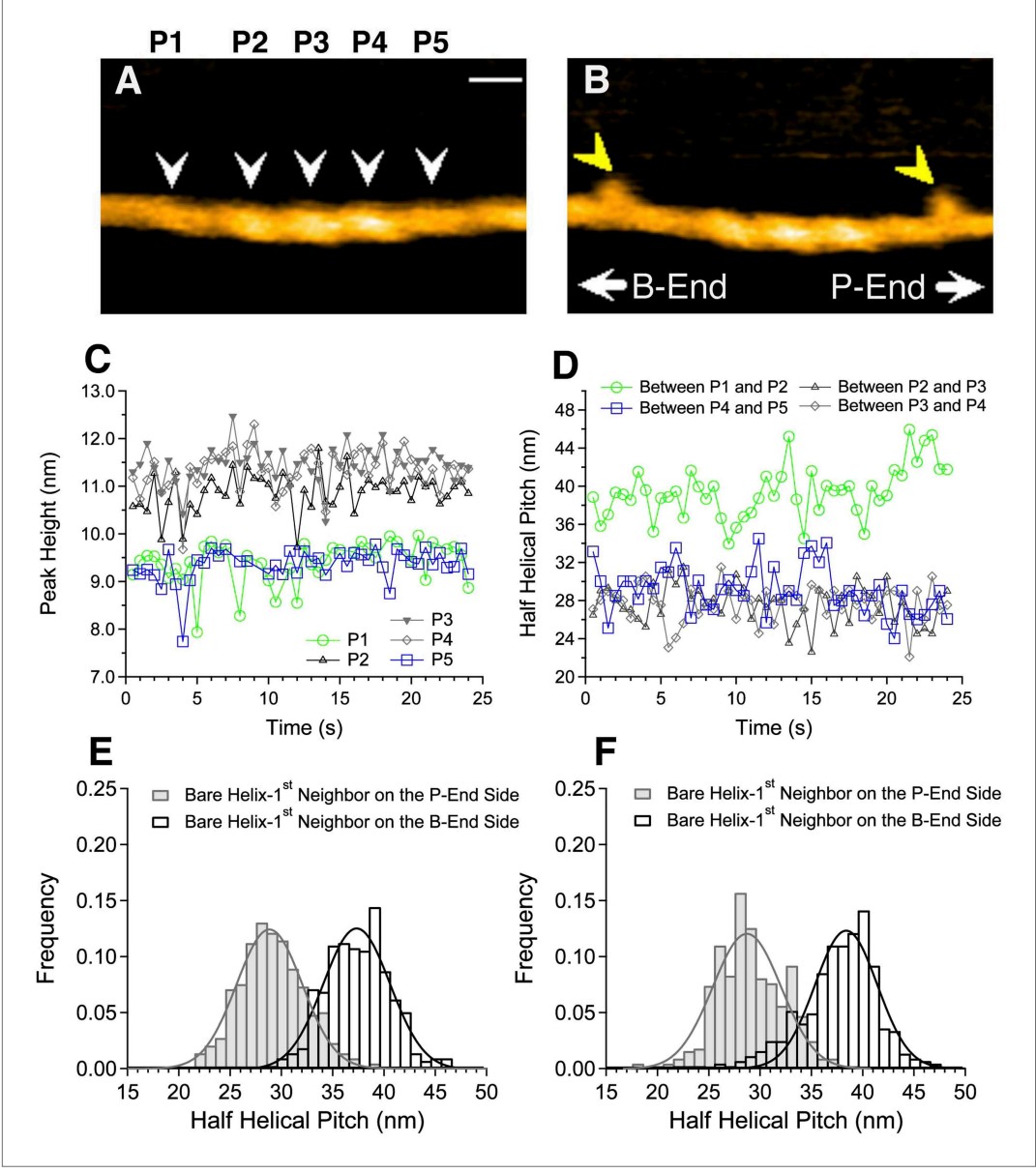

**Figure 2**. Asymmetric structure of bare actin zones neighboring a cofilin cluster. (**A** and **B**) HS-AFM image of a short cofilin cluster transiently associating with two S1 molecules (yellow arrowheads), which persisted for ~1 s, enabling identification of the filament polarity (**B**). Measurements were made in F buffer containing 20 nM S1, 75 nM cofilin, 1 mM ADP, and 0.1 mM ATP. Bar: 25 nm; Z-scale: 0–12 nm. (**C** and **D**) Time-dependent changes in the heights of the indicated peaks (white arrowheads) and half helical pitches between the indicated peaks. (**E** and **F**) Histograms of the lengths of the half helical pitches of bare actin segments immediately neighboring cofilin clusters. Filaments were incubated in F buffer containing 1 mM ADP and 0.1 mM ATP for 5 min before observation (**E**) as in (**A** and **B**) or were incubated in F buffer containing 1 mM ADP for 30 min before observation (**F**). Pitches of the half helices of the first immediate neighbor on each side of cofilin clusters were measured. These values, together with those for the second neighbors, are summarized in *Table 1*.

microscopic observation described above, we were unable to detect significant changes in peak heights and half helical pitches, such as those observed around cofilin clusters, around those singly bound cofilin-rod molecules (*Figure 3E,F*).

These results suggest that a single cofilin molecule cannot induce conformational changes, either symmetric or asymmetric, that involve detectable changes in helical pitch when it binds to bare sections of the filament.

**Table 1.** Peak heights and lengths of half helical pitches in bare actin segments neighboring cofilin clusters

| | 1 mM ADP + 0.1 mM ATP | | 1 mM ADP | |
|---|---|---|---|---|
| | Peak height (nm) | Half helix (nm) | Peak height (nm) | Half helix (nm) |
| First neighbor on the P-end side | 9.2 ± 1.0 | 28.8 ± 4.5 | 8.9 ± 1.0 | 28.7 ± 4.7 |
| Second neighbor on the P-end side | 9.1 ± 1.0 | 36.5 ± 4.1 | 8.8 ± 1.1 | 36.9 ± 4.9 |
| First neighbor on the B-end side | 9.0 ± 0.9 | 37.3 ± 4.6 | 8.7 ± 0.9 | 38.4 ± 4.3 |
| Second neighbor on the B-end side | 9.2 ± 0.6 | 35.4 ± 3.9 | 8.7 ± 0.8 | 36.7 ± 4.2 |

Actin filaments were incubated in F buffer containing 1 mM ADP and 0.1 mM ATP for 5 min or in F buffer containing 1 mM ADP for 30 min prior to the addition of cofilin. Filaments under the latter condition were shorter than those under the former condition and were apparently in the process of spontaneous depolymerization.
Each mean and SD were calculated from 423 to 446 data.

## Growth of the cofilin clusters

Real time AFM imaging taken at 1.5 or 2 frames/s enabled us to follow the growth of individual cofilin clusters along actin filaments. In the case illustrated in *Figure 4A* and *Video 3*, observed in F buffer containing 1 mM ADP and 0.1 mM ATP, P4 was already tall when imaging started (~10.5 nm, average of t = 0–5 s), while P3 had an intermediate height (~9.5 nm), and P1 and P2 were normal (~8.5 nm) (*Figure 4A*, middle). We interpreted this to mean that the initial cofilin cluster extended beyond P4 but ended near P3. P3 gradually became taller and plateaued beyond ~25 s. Thereafter, P2 started to rise at ~15 s and plateaued at ~63 s, whereas P1 started to rise at ~60 s. The distance between P2 and P3 rapidly shortened to ~27 nm between 0 and 15 s, before P2 started to rise, while the distance between P1 and P2 shortened between 30 and 40 s, before P1 started to rise (*Figure 4A*, bottom). This sequence of events implies gradual growth of this cofilin cluster into the neighboring bare zone on the pointed-end side where the helical pitch was shortened.

Cofilin binds preferentially to actin subunits carrying ADP, compared to those carrying ATP or ADP and Pi (*Carlier et al., 1997*; *Blanchoin and Pollard, 1999*). Since actin filaments in F buffer containing ATP should have ATP–actin caps near the barbed ends, and since subunits on the pointed-end side tend to carry ADP only due to ATP hydrolysis and Pi release (*Carlier, 1990*), the directional growth of cofilin clusters toward the pointed-end might reflect the asymmetric distribution of actin subunits with different nucleotides. To test this possibility, we prepared two different types of actin filaments with homogenous nucleotide states along the lengths. In the first case, filaments polymerized in buffer containing 1 mM ATP were incubated for 30 min in buffer containing 1 mM ADP, which is much longer than the 350 s required to hydrolyze ATP and release the resultant Pi from half of the polymerizing ATP–actin molecules (*Melki et al., 1996*), so that most of the actin subunits should carry ADP only. In the second case, filaments polymerized in buffer containing 1 mM ATP were incubated for 10 min in buffer containing 1 mM ADP and 10 mM Pi. Under this condition, most of the actin subunits should carry ADP and Pi, considering a $K_d$ of 1.5 mM for Pi (*Carlier and Pantaloni, 1988*). HS-AFM observations after the addition of 20 nM S1 and 75 nM cofilin or 150 nM S1 and 900 nM cofilin demonstrated that, in both cases, the growth of cofilin clusters was primarily to the pointed-end direction (*Figure 4B* and *Video 4*, and *Figure 4C* and *Video 5*). Results of a large number of observations are compiled in *Figure 5*.

The rate of formation and growth of cofilin clusters depended on the nucleotide state of actin subunits as well as the concentration of cofilin. In most experiments we used 75 nM cofilin since we were able to observe de novo formation of clusters and tractable growth, after mixing and settling of the system. When 1 μM cofilin was added and mixed, we were unable to find bare zones of actin filaments. When actin filaments were incubated with 1 mM ADP and 10 mM Pi, much higher concentrations of cofilin (e.g., 900 nM) was needed to induce cofilin clusters. The growth rates of cofilin clusters were within tractable range even in the presence of 1 μM cofilin, when 10 mM Pi was present (*Figure 4C*). These results are consistent with the weaker affinity of cofilin for actin subunits carrying ADP and Pi (*Carlier et al., 1997*; *Blanchoin and Pollard, 1999*).

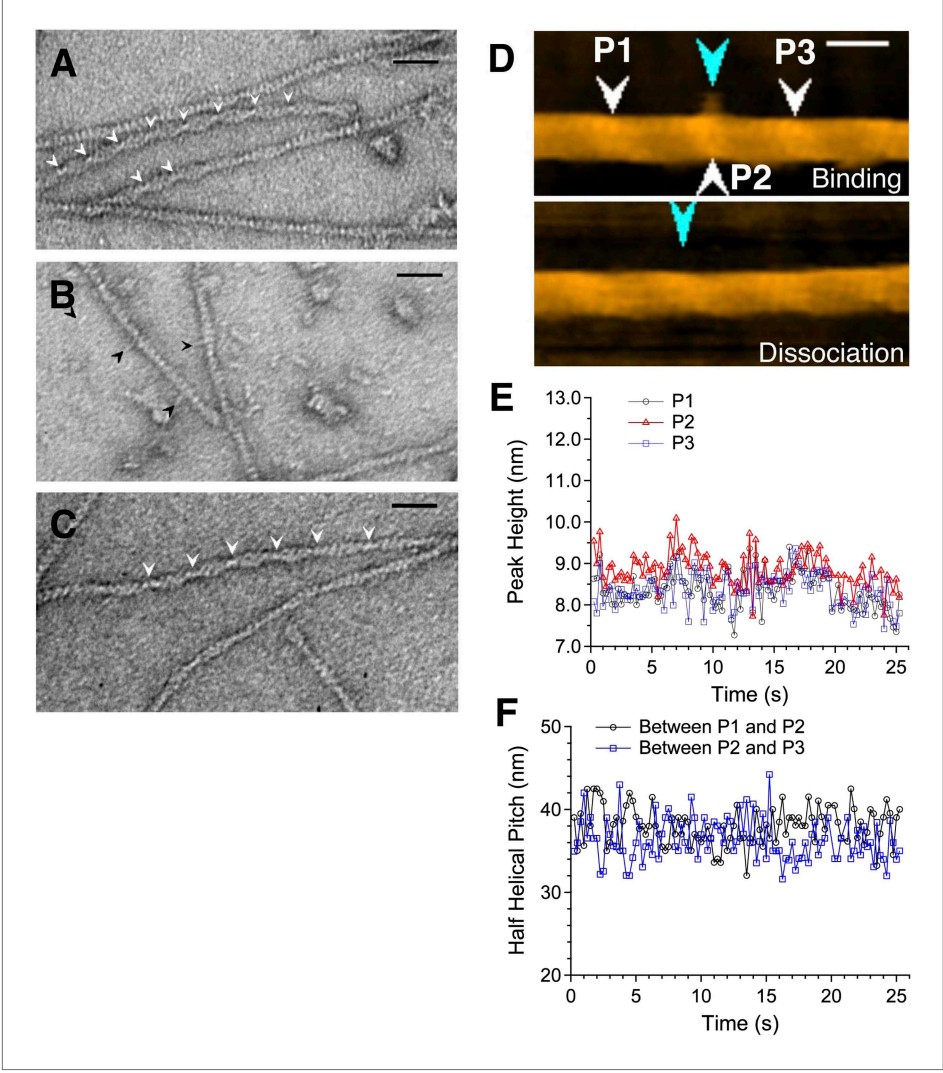

**Figure 3**. Actin filaments with bound cofilin or cofilin-rod fusion protein. (**A–C**) are electron micrographs of negatively stained samples, and (**D**) is a HS-AFM image of a sample similar to that shown in (**B**). (**A**) Actin filaments bound with cofilin-rod. Arrowheads show crossover points in clusters of cofilin-rod. The rod portions of the fusion proteins are not readily visible, which may be due to alignment of the rods along the cofilin clusters. Severing activity and stoichiometric binding of cofilin-rod to actin filaments were confirmed by HS-AFM (**Video 1**) and co-sedimentation assays (**Figure 3—figure supplement 3**), respectively. (**B**) Cofilin-rod molecules sparsely bound to actin filaments, identified by the rod-like structures (black arrowheads). (**C**) Actin filaments with bound cofilin molecules (without rod fusion). Arrowheads show crossover points in clusters. Actin filaments and cofilin or cofilin-rod were mixed at a 2:1 (**A** and **C**) or 1:1 (**B**) molar ratio in F buffer containing 1 mM ATP. Bars: 25 nm. (**D**) HS-AFM image of an actin filament and an apparently singly bound cofilin-rod molecule (blue arrowhead in the upper image) near P2 (white arrowhead). Conditions: F buffer containing 1 mM ATP and 75 nM cofilin-rod (without His-tag). Bar: 25 nm. See **Video 2**. (**E**) shows heights of the three peaks and (**F**) shows spacing between them.

The following figure supplements are available for figure 3:

**Figure supplement 1**. Co-sedimentation of cofilin (with or without His-tag) with actin filaments.

**Figure supplement 2**. Actin binding curves of cofilin and cofilin without His tag.

**Figure supplement 3**. Co-sedimentation of cofilin-rod with (+) and without (−) His-tag.

**Figure supplement 4**. Representative still images from **Video 1**, demonstrating cluster formation and severing function of cofilin-rod without His-tag.

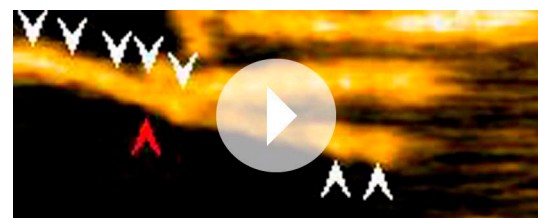

**Video 1**. Cluster formation and severing functions of cofilin-rod. Conditions: F buffer containing 1 mM ATP, and 300 nM cofilin-rod without His-tag, imaging rate: 2 frames/s, and playing rate: 5 frames/s. White arrowheads indicate clusters of cofilin-rod, and red and blue arrowheads show severing points inside a cluster and in a bare half helix immediately neighboring a cluster, respectively. Z-scale was 0–12 nm. Related to *Figure 3—figure supplement 4*.

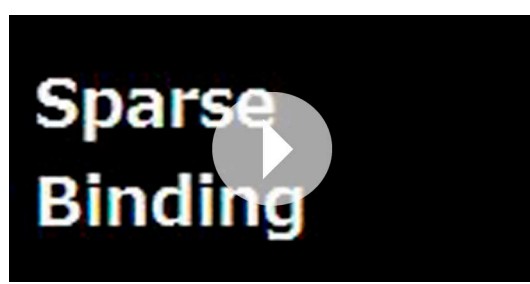

**Video 2**. Sparse binding of individual cofilin-rod molecules to actin filaments. Conditions: F buffer containing 1 mM ATP and 75 nM cofilin-rod without His-tag, imaging rate: 4 frames/s, and playing rate: 5 frames/s. Transient binding of cofilin-rod to and dissociation from an actin filament was followed for approximately 90.5 s, shown by the presence and absence of a blue arrowhead. No severing was observed in all four similar cases of successful imaging of sparse binding of cofilin-rod to actin filaments. Z-scale was 0–12 nm. Related to *Figure 3D*.

## Severing of filaments by cofilin

Cofilin is an actin filament severing protein (reviewed by *Pollard (2000)*), and although our experiments were performed at pH 6.8, which should suppress severing activity (*Yonezawa et al., 1985*; *Hawkins et al., 1993*; *Pavlov et al., 2006*), we observed frequent severing of the filaments near or within short cofilin clusters (*Figure 6A–C*, *Videos 6–10* and *Figure 6—figure supplements 1 and 2*). Because a previous study by *Adrianantoandro and Pollard (2006)* found that severing activity is highest at 10 nM of human cofilin and that this activity sharply declines above and below 10 nM, we observed severing events at several different cofilin concentrations, including 10 nM. In the presence of 10 or 20 nM cofilin, severing was very infrequent (*Video 11* and *Figure 6—figure supplement 3*). Severing was often observed between 40 (*Video 12* and *Figure 6—figure supplement 4*) and 200 nM cofilin. In the presence of 650 nM cofilin, many of the filaments were fully decorated with cofilin when observations started, and severing was observed exclusively near the ends of the long clusters (*Video 12* and *Figure 6—figure supplement 4*). When the cofilin concentration was increased to 1 μM, all the filaments were fully decorated with cofilin, and severing was infrequent along those fully decorated filaments (*Video 12*). Thus, we also observed that cofilin's severing activity was highest in the medium concentration range (i.e., 40–200 nM), but this concentration is at least several-fold higher than that reported by *Adrianantoandro and Pollard (2006)*. These two studies used the same pair of proteins (i.e., skeletal muscle actin and human cofilin), and we can only speculate that this discrepancy is due to differences in experimental conditions.

We next quantitatively analyzed the severing positions of the filaments relative to the short cofilin clusters in the presence of 40–200 nM cofilin (*Figure 6D*). Approximately 60% of severing events occurred inside the cofilin clusters, mostly within half helices neighboring bare zones. Severing also occurred in half helices in bare zones neighboring a cofilin cluster, which accounted for approximately 40% of severing events. Overall, ~80% of the severing events occurred within one-half of a helix on either side of the boundary between a bare zone and a cofilin cluster. Severing in bare zones far from cofilin clusters was very rare.

In the presence of 40 nM cofilin, the lowest concentration we used for quantitative severing assay, we observed a total of 22 cases of severing events in 31 actin filaments. Among those, 18 cases occurred in half helices immediately neighboring the boundary between a bare zone and a cofilin cluster, even though cluster formation was relatively rare at this concentration of cofilin. Three cases occurred in 'far' bare zones more than half a helix away from the boundary, and one occurred in an 'inner' cofilin cluster more than half a helix away from the boundary (*Video 11*).

Based on these observations, we conclude that severing under the present experimental condition preferentially occurs near the boundary between a bare zone and a cofilin cluster, but not necessarily at the boundary. Considering the difference in spatial resolution, this conclusion is also consistent with

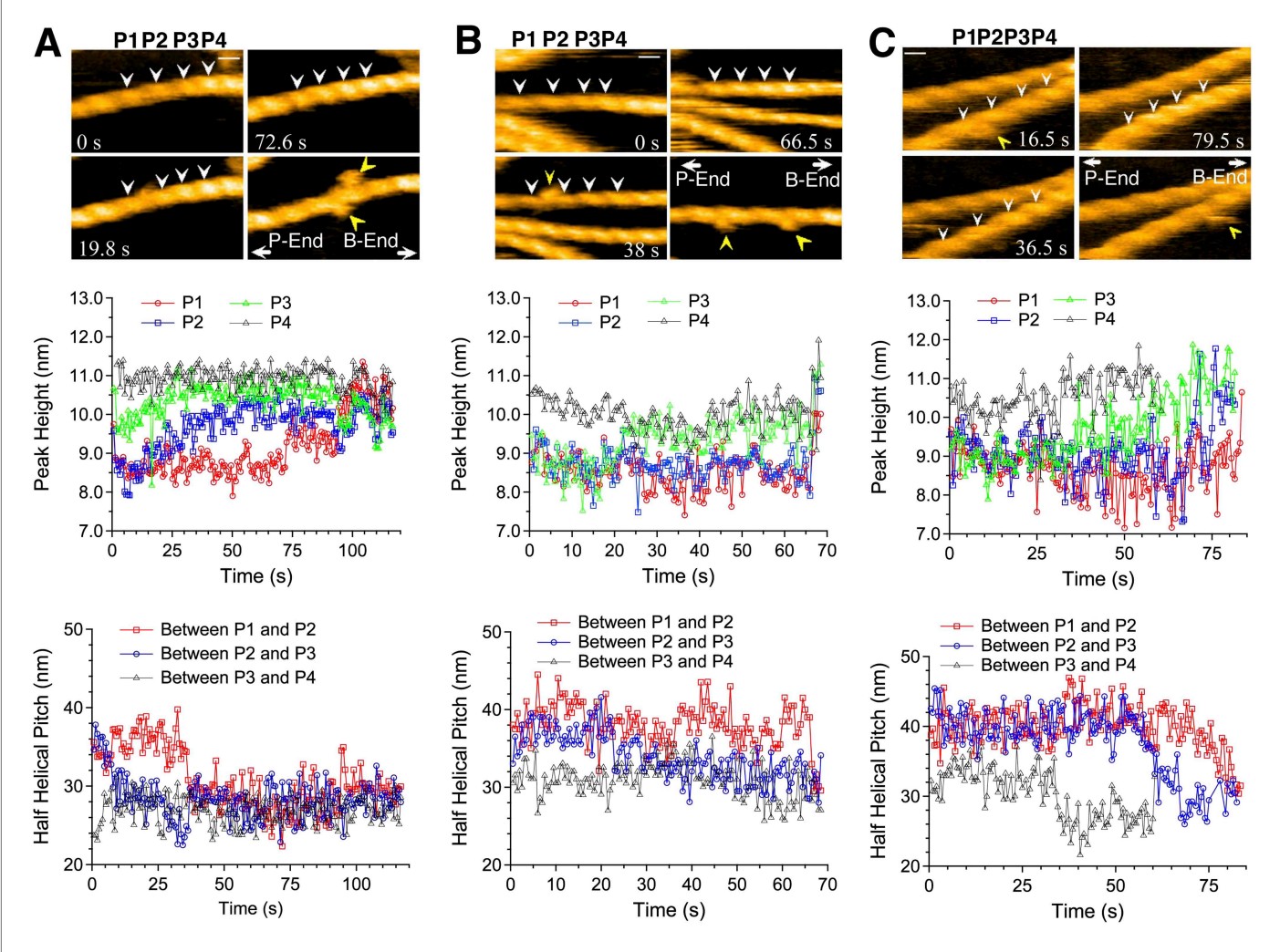

**Figure 4.** Growth of cofilin clusters along actin filaments. Growth of cofilin clusters along actin filaments in F buffer containing 1 mM ADP and 0.1 mM ATP (**A**), along actin filaments carrying ADP, prepared by incubating filaments in F buffer containing 1 mM ADP, hexokinase, and glucose at room temperature for 30 min (**B**), and along actin filaments carrying ADP and Pi, prepared by incubating filaments in 1 mM ADP and 10 mM Pi for 10 min at room temperature (**C**). The concentrations of cofilin were 75 nM (**A** and **B**) or 900 nM (**C**) and those of S1 were 20 nM (**A** and **B**) or 150 nM (**C**). Each panel consists of four sequential snapshots (top), a figure showing the heights of the indicated peaks (middle), and the half helical pitches between the indicated peaks (bottom). Yellow arrowheads show the transient association of S1. Note that P1 in (**A**) and P3 in (**B**) rose in two substeps. Bars: 25 nm; Z-scale: 0–12 nm.

that made by *Suarez et al. (2011)*, who used fluorescence microscopy and concluded that severing occurs at the boundary between cofilin clusters and bare zones.

## Discussion

Using HS-AFM to visualize wet samples with high-spatial and -temporal resolution, we were able to create an image of conformational changes in actin filaments induced by cofilin binding, including asymmetric changes in the helical twist on either side of a bound cofilin cluster, growth of the clusters toward the pointed-end, and the severing of filaments near the ends of cofilin clusters.

### Unidirectional propagation of the cofilin-induced supertwisted conformation

The short helical pitch in the cofilin clusters was propagated to the immediately neighboring bare zone on the pointed-end side of the cluster. Since our current analysis method only measures heights and

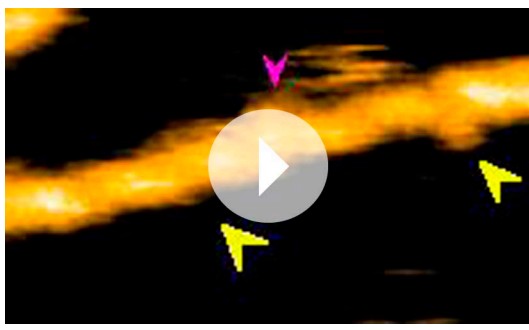

**Video 3**. Growth of a cofilin cluster toward the pointed end of a filament in F buffer containing 1 mM ADP, 0.1 mM ATP, 20 nM S1, and 75 nM cofilin. Imaged at 2 frames/s and played at 5 frames/s. White arrowheads show growth of the cofilin cluster, and yellow and magenta arrowheads show binding of S1. Magenta arrowheads indicate S1 molecules whose binding angle could not be determined, either for geometric reasons (i.e., binding on the upper face of the filament) or because the binding was too short-lived. Z-scale was 0–12 nm. For magnifications and polarity of the analyzed filaments, refer to *Figure 4* in the main text. Related to *Figure 4A*.

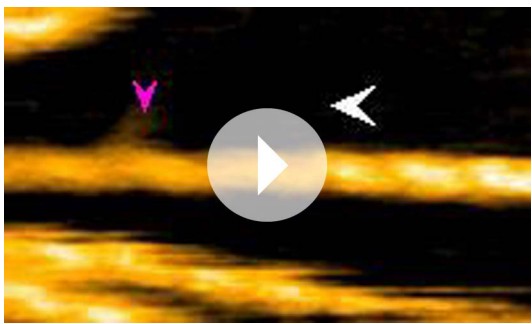

**Video 4**. Growth of a cofilin cluster toward the pointed end of a filament in F buffer containing 1 mM ADP, 20 nM S1, and 75 nM cofilin. Imaged at 2 frames/s and played at 5 frames/s. For color codes of arrowheads, see the legend to *Video 3*. Z-scale was 0–12 nm. For magnifications and polarity of the analyzed filaments, refer to *Figure 4* in the main text. Related to *Figure 4B*.

distances between peaks, we were unable to examine if the supertwisted structure returns to a normal state abruptly or gradually. Also, we were unable to determine the positions of the ends of cofilin clusters precisely above the spatial resolution of half helices, making it difficult to estimate accurately how far the supertwisting conformational changes propagate into neighboring bare zones. Nonetheless, the fact that the pitches of the neighboring half helices second from the cofilin clusters were nearly normal suggests that the effect does not propagate longer than one-half of a helix.

The neighboring bare zone on the barbed-end side of the cluster had slightly longer helical pitch than the control. Although a *t*-test indicated that the untwisting was statistically significant, we are not certain if it was an active process. If interactions between actin filaments and the lipid surface produced resistive force against local rotation of the filament, supertwisting in the cluster would induce passive, compensating untwisting of the nearby helix. If that is the case, the fact that actin filaments are not necessarily free to rotate in vivo suggests similar passive untwisting may occur in vivo as well. In any case, it is noteworthy that cofilin clusters induced distinctly asymmetric cooperative conformational changes in neighboring bare zones. Propagation of the supertwisted conformation from cofilin clusters to neighbor bare zones had been suggested by image analysis of electron micrographs of cofilin–actin complexes (*Galkin et al., 2001*).

## Growth of cofilin clusters is also unidirectional and independent on actin-bound nucleotides

Our observation that the cofilin clusters grew in the pointed-end direction supports the conclusion of *Galkin et al. (2001)* that cofilin preferentially binds to the supertwisted segments of actin subunits. This tendency of cofilin clusters to grow unidirectionally was unaffected when most of the filament subunits carried ADP or ADP+Pi, demonstrating that the directional growth of cofilin clusters does not depend on a gradient of different nucleotide states on actin subunits within each filament. Instead, it presumably depends on asymmetric cofilin-induced conformational changes in the actin filaments, which is an intrinsic property of the polar structure of the actin filaments.

The rise of peaks and shortening of half helical pitches that accompanied the growth of cofilin clusters were sometimes rapid, but at other times slow and gradual (*Figure 4*). If we assume that the rise was due to simple addition of cofilin molecules at the two binding sites closest to the crossover point, and if the cofilin clusters grow in perfect synchrony along two strands of the double helix, then the crossover points should rise abruptly in one step. If, on the other hand, there is a time-lag between cofilin bindings at the two critical binding sites, due to delayed growth along one strand, for example, the rise would occur in two smaller discrete substeps. Indeed, in many cases, the rise of peaks appeared

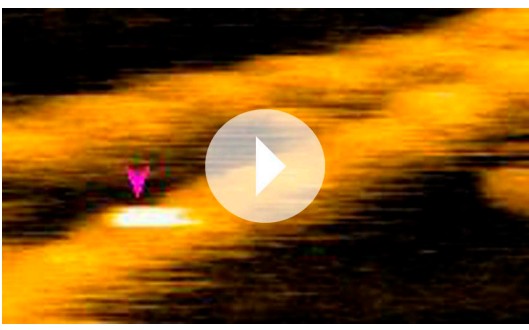

**Video 5**. Growth of a cofilin cluster toward the pointed end of a filament in F buffer containing 1 mM ADP, 10 mM Pi, 150 nM S1, and 900 nM cofilin (without His-tag). For color codes of arrowheads, see the legend to **Video 3**. For magnifications and polarity of the analyzed filaments, refer to **Figure 4** in the main text. Under this condition, binding of S1 was so short-lived that the tilted binding was not obvious in some cases (magenta). The S1 molecule indicated by a double magenta arrowhead appears to tilt in the direction opposite to other S1 molecules indicated by yellow arrowhead, and we speculate that this is because this S1 molecule was not stably bound to the filament when imaged. Imaged at 2 frames/s and played at 3 frames/s. Z-scale was 0–12 nm. Related to **Figure 4C**.

to occur in two discrete substeps (**Figure 4**), supporting this possibility. In other cases, however, the peaks rose gradually over the course of 40 s (e.g., P2 in **Figure 4A**), although we have no mechanistic explanation for those events.

In certain cases the next peak (P1 in the case shown in **Figure 4A**) only rose when the previous peak (P2) had reached a plateau, whereas in other cases the new peak (P2) started to rise while the previous one (P3) was still rising. This latter case is more consistent with the possibility that cofilin clusters do not necessarily grow in concert along the two strands, and the delay may sometimes exceed one-half of a helix.

Another uncertainty is whether the new cofilin molecule always binds to the vacant binding site immediately neighboring the cluster. If the unidirectional cluster growth to the pointed-end is driven by the shortened helical pitch on the pointed-end side of the cluster, the new molecule is likely to bind any of the vacant binding sites in the supertwisted, apparently, bare zone on the pointed-end side of the cluster. This view is consistent with the result of recent high-resolution single molecule fluorescence microscopic assays that showed that free cofilin molecules prefer to bind within 65 nm of an already bound cofilin molecule, but not necessarily to the site immediately neighboring the already bound molecule (**Hayakawa et al., 2014**). We thus speculate that there are vacant binding sites behind the advancing front of cofilin clusters, which are eventually filled by other cofilin molecules to form tight clusters, and that the advances are not necessarily in concert between the two strands of the double helix.

## Severing

In the presence of cofilin, actin filaments are frequently severed at or near the boundary between cofilin clusters and bare zones, at least when examined at the spatial resolution of fluorescence microscopy (**Suarez et al., 2011**). That observation is consistent with the idea that severing preferentially occurs at sites of structural discontinuity, such as the boundary between supertwisted and normal helical pitches (**Michelot et al., 2007**; **De La Cruz, 2009**). An alternative view is that cofilin binding weakens longitudinal contacts between actin subunits within cofilin clusters and also in neighboring bare zones, but because bound cofilin bridges two actin subunits, strengthening interactions between them, severing occurs in nearby supertwisted bare zones (**Galkin et al., 2001**; **Bobkov et al., 2002**). Roughly 40% of the severing events in our AFM observations occurred at the boundary between a cluster and a bare zone or within the neighboring bare zone, which may be explained by these hypotheses. However, more than half of the severing events occurred within the cofilin clusters, albeit close to the cluster ends. One plausible explanation is that the scattered unoccupied cofilin binding sites near the ends of the cluster, discussed above, may be easy to break due to the lack of bridging cofilin molecules. Alternatively, if growth of cofilin clusters on one of the two filament strands lags the growth on the other, conformational stress may develop between the two strands, leading to breaks in the filament near the ends of cofilin clusters.

## How many bound cofilin molecules are required for severing and cluster growth?

Previous biochemical (**Andrianantoandro and Pollard, 2006**) and simulation studies (**De La Cruz, 2005**; **De La Cruz, 2009**) suggested that one or a few bound cofilin molecules are sufficient to sever actin filaments. However, our AFM observations are more consistent with the view that efficient

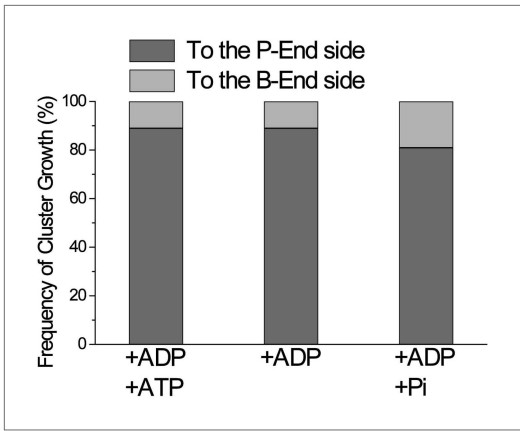

**Figure 5**. Directional preference of the growth of cofilin clusters. The growth of cofilin clusters was observed under three buffer conditions: in the presence of 1 mM ADP and 0.1 mM ATP (+ADP +ATP); 1 mM ADP (+ADP) and 1 mM ADP and 10 mM Pi (+ADP +Pi), as in *Figure 4*. Growth of a cluster by one-half helix was counted as one growth event. The total number of observed growth events was 37, 46, and 188 for each condition. We speculate that at least some of the cluster growth events in the barbed-end direction were actually growth in the preferred direction from invisibly small clusters on the barbed side of a visible cluster.

severing requires a cofilin cluster longer than one-half of a helix, since most of the severing events were observed within or very near cofilin clusters that were recognizable in our AFM images. The need of contiguously bound cofilin molecules for efficient severing is consistent with our earlier mutant analysis. Within filaments in solution, D11Q mutant actin subunits rapidly exchange bound ADP for ATP in solution. Consequently, most subunits within D11Q actin filaments have ATP bound, reducing their affinity for cofilin and protecting them from severing. Interestingly, cofilin was able to bind to copolymers of D11Q and wild-type actins, but severing was inefficient (*Umeki et al., 2012*), suggesting that contiguous clusters of cofilin-bound actin subunits are necessary for efficient severing activity.

A related question is how many bound cofilin molecules are required to induce cooperative supertwisting conformational changes in actin filaments? Answering this question is technically difficult when the imaging power is not sufficient to see individual bound cofilin molecules, but our observations using a cofilin-rod fusion protein showed that one is not enough. Consistent with this view, previous simulation studies suggested that two cofilin molecules bound close to one another along an actin filament serve as a nucleus to initiate cooperative binding of cofilin to form a cluster (*Ressad et al., 1998*; *Blanchoin and Pollard, 1999*). Future high-resolution HS-AFM studies using cofilin fused with a rod or some other structural marker to make it visible with AFM will directly test those hypotheses.

Even if a singly bound cofilin molecule cannot induce supertwisting of the helix or initiate cluster growth, it does not exclude the possibility that single bound cofilin molecules induce subtler, perhaps longer range, cooperative conformational changes, such as those indirectly detected through biophysical measurements (*Dedova et al., 2004*; *Prochniewicz et al., 2005*; *Bobkov et al., 2006*). Apparently cofilin is able to induce at least two distinct types of cooperative conformational changes: one that requires cluster formation, involves ~25% supertwisting of the helix, and propagates over half a helix toward the pointed-end of the filament; and a second type that singly bound cofilin molecules can induce which involves relatively subtle conformational changes, and propagates much longer. Simulation studies by *De La Cruz and Sept (2010)* suggest that there are two distinct states of cofilin–actin complexes, which may be correlated with the two types of cooperative conformational changes we propose here.

## Physiological implications of cofilin-induced unidirectional cooperative conformational changes in actin filaments

Cooperativity in the binding of cofilin to actin filaments could have multiple physiological implications. Generally speaking, cooperativity would amplify small changes in the input (concentration of active cofilin) to a larger difference in output (cluster formation and severing). In addition, the cellular concentration of cofilin is lower than that of polymerized actin, and cooperativity would be a useful means of disrupting selected filaments under those conditions (*Pollard et al., 2000*).

Second, we propose that the propagation of cofilin-induced conformational changes into neighboring cofilin-unbound zones of actin filaments would give cofilin an advantage in competition with other ABPs, once a small cofilin cluster is established as a foothold. For example, cofilin is implicated in severing and depolymerization of aged actin filaments in lamellipodia. However, those actin filaments are often bound with tropomyosin (*Gunning et al., 2008*), which inhibits binding of cofilin and protects the filaments from cofilin's severing and depolymerizing activities (*Bernstein and Bamburg, 1982*; *Ono and Ono, 2002*). If cofilin forms a small cluster at a vacant site on an actin filament that is

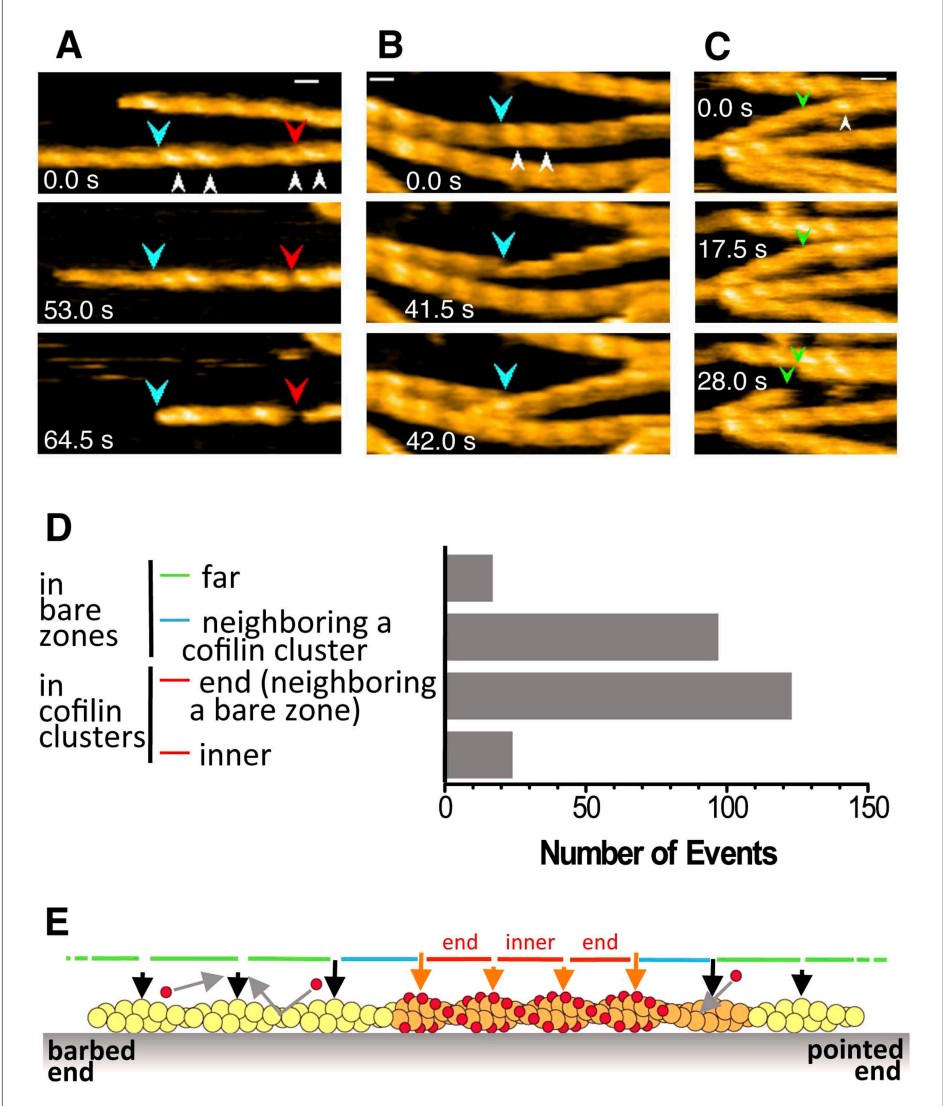

**Figure 6**. Severing of actin filaments near cofilin clusters. (**A**–**C**) Typical cases of filament severing (red arrowheads) within or near cofilin clusters (white arrowheads). The observation buffers were F-buffer containing 1 mM ADP (**A** and **C**) and 1 mM ATP (**B**). Concentration of cofilin was 40 nM (**A** and **C**) and 75 nM. The first break in (**A**) was inside a cluster, while the second was at or near the junction between a bare zone and a cluster. Red, blue, and green arrowheads show severing points within cofilin clusters, in bare zone close to cofilin clusters, and in bare zones more than half a helix away from a cofilin cluster, while white arrowheads show cofilin clusters. Bars: 25 nm; Z-scale: 0–12 nm. See **Videos 6–8**. (**D**) Classification of severing sites into four categories: (1) in 'far' bare zone (between a tall and a short black arrow or between two short black arrows, indicated by green bars in (**E**)); (2) in bare zone half helices immediately neighboring a cofilin cluster (between a tall black and a tall orange arrow, indicated by blue bars in (**E**)); (3) in 'end' cofilin cluster half helices immediately neighboring bare zones (between a tall and a short orange arrow, indicated by red bars in (**E**)); and (4) in 'inner' cofilin cluster half helices (between two short orange arrows, indicated by a red bar in (**E**)). Comparison of the last two categories demonstrates that severing within cofilin clusters occurs preferentially near the ends. Note, however, that this comparison does not necessarily show a quantitative difference in the susceptibility to severing between end and inner half helices, since the number of end and inner helices examined are not the same. (**E**) A schematic summary of the proposed distributions of bound cofilin molecules (red spheres), segments of normal (yellow) and shortened (orange) helical pitch, and normal (black arrows) and tall (orange arrows) crossover points. Free cofilin molecules tend to bind to the supertwisted bare zone on the pointed-end side of the cluster (gray arrows), driving the growth of the cluster in the pointed-end direction. This is most certainly an oversimplification, ignoring a number of complex issues, some of which are discussed in the main text.

*Figure 6. Continued on next page*

*Figure 6. Continued*

The following figure supplements are available for figure 6:

**Figure supplement 1**. Representative still images from *Video 9*, demonstrating severing of actin filaments by cofilin with His-tag.

**Figure supplement 2**. Representative still images from *Video 10*, showing severing of actin filaments by cofilin without His-tag.

**Figure supplement 3**. Representative still images from *Video 11*, showing severing of actin filaments in the presence and absence of low concentration of cofilin.

**Figure supplement 4**. Representative still images from *Video 12*, showing severing in actin filaments decorated with high concentrations of cofilin.

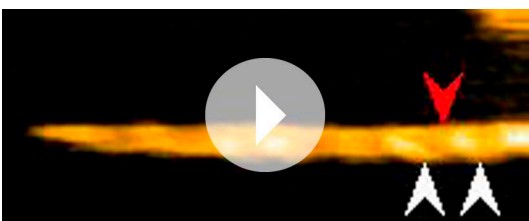

**Video 6**. Severing of actin filaments in a cofilin cluster. Conditions: F buffer containing 1 mM ADP and 40 nM cofilin. Severing of actin filaments occurred within a cofilin cluster (at 53 s) and then at or near the boundary between a bare zone and another cofilin cluster (at 64.5 s). Imaged at 2 frames/s and played at 5 frames/s. Red, blue, and green arrowheads indicate severing in half helices in cofilin clusters, in a bare half helix immediately neighboring a cofilin cluster, and in bare zones more than half a helix away from cofilin clusters, respectively. White arrowheads: cofilin clusters. Z-scale was 0–12 nm. Related to *Figure 6A*.

otherwise decorated and protected by tropomyosin, the cluster would induce, or apply conformational stress to induce, supertwisting on the neighboring tropomyosin-bound segment on the pointed-end side. This would accelerate dissociation of tropomyosin, resulting in faster growth of the cofilin cluster than when cluster growth needed to wait for spontaneous dissociation of neighboring tropomyosin molecules. Furthermore, that cluster growth is directed only to the pointed-end may be beneficial in selective disassembly of aged actin filaments.

Within cells, formin remains bound to the barbed-end when it catalyzes filament elongation. This causes relative rotation between the formin molecule and the filament (*Mizuno et al., 2011*). Thus, if the formin molecule and the filament are not free to rotate, the filament will be untwisted, and if cofilin has a lower affinity for untwisted actin filaments, rapidly polymerizing actin filaments in a formin-dependent manner will be protected from severing by cofilin (*Mizuno and Watanabe, 2012*). It should be noted, however, that formin bound to the barbed-end of an actin filament also allosterically changes the structure of the filament (*Bugyi et al., 2006*), which may interfere with the interactions of the filament with cofilin, independent of mechanically forced untwisting of the filament helix.

*Sharma et al. (2012)* discovered that the untwisted conformation of actin filaments induced by a drebrin N-terminal fragment propagates to neighboring bare zones; however, their findings seem to indicate that this propagation is in both directions from the drebrin clusters, though the authors did not address that point. It thus appears that there are multiple forms of ABP-induced cooperative conformational changes to actin filaments that propagate into bare zones, which implies there are also multiple physiological functions for such cooperative conformational changes to actin filaments.

## Materials and methods

### Proteins

cDNA encoding human cofilin 1 was amplified from a human cDNA library using PCR with primers 5′-ggtaccatggcctccggtgt and 5′-tctagacaaaggcttgccctcca. After confirmation of its sequence, the amplified DNA fragment was subcloned into pColdI expression vector (Takara Bio, Otsu, Japan) at the *Xba*I and *Kpn*I sites. The pColdI vector had been modified to contain a TEV cleavage site between the His tag and the multi-cloning sites, so that the amino acid sequence near the N-terminus was

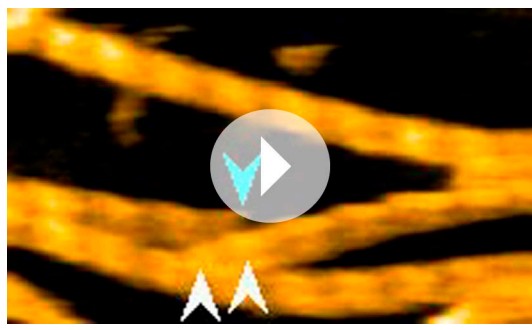

**Video 7**. Severing of actin filaments at or near a boundary between a bare zone and a cofilin cluster. Conditions: F buffer containing 1 mM ATP and 75 nM cofilin. Severing occurred at 42 s. Imaged at 2 frames/s and played at 5 frames/s. For color codes of the arrowheads, see the legend to **Video 6**. Z-scale was 0–12 nm. Related to **Figure 6B**.

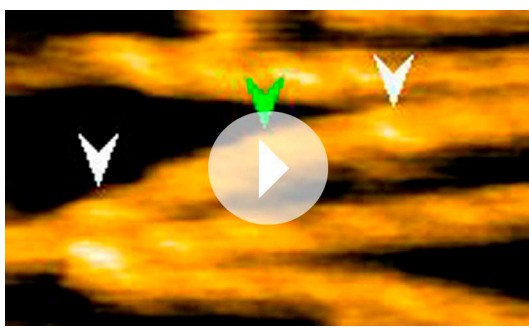

**Video 8**. Severing of actin filaments in a bare zone more than one half helix away from a cofilin cluster. Conditions: F buffer containing 1 mM ADP and 40 nM cofilin. Severing occurred at 28 s. Imaged at 2 frames/s and played at 5 frames/s. For color codes of the arrowheads, see the legend to **Video 6**. Z-scale was 0–12 nm. Related to **Figure 6C**.

MNHKVHHHHHHIEGRHM*ENLYFQG*TM ASGVAVS… (italics indicate the TEV cleavage site).

The cofilin-rod fusion gene was constructed by cloning the cDNA encoding the first half of the *Dictyostelium* α-actinin rod downstream of the cofilin gene in pColdITEV. The amino acid sequence at the junction of the two proteins was …SAVISLEGKP*L*EQTKSDYLKRA…, and the C terminal sequence was …QKIEDSLV*SR* (italics show extra amino acid residues derived from recognition sites for restriction enzymes). The first half of the *Dictyostelium* α-actinin rod, corresponding to amino acid residues 265–505 of the parent molecule, forms a monomeric12-nm long rod-like structure (*Yan et al., 1993*) and has been used as an artificial lever arm of myosin motors (*Anson et al., 1996*).

The proteins were expressed in *Escherichia coli,* purified using Ni-NTA affinity chromatography, and dialyzed against a buffer containing 10 mM HEPES, pH 7.4, 50 mM KCl, 0.1 mM DTT, and 0.01% $NaN_3$ overnight at 4°C. After concentrating with a centrifugal concentrator (Amicon Ultra 4), aliquots were snap-frozen in liquid nitrogen and stored at −80°C. Unless otherwise stated, the experiments used those His-tagged cofilin or cofilin-rod proteins. However, we repeated some key experiments after removing the His-tag by treatments with TEV protease and obtained qualitatively similar results. These experiments included asymmetric conformational changes of actin filaments on either side of a cofilin cluster, unidirectional growth of cofilin clusters in the pointed-end direction, and frequent severing of filaments near the boundary between a bare zone and a cofilin cluster. Cofilin with and without His-tag bound to actin filaments at a 1:1 molar ratio with a similar affinity (*Figure 3—figure supplements 1 and 2*). Cofilin-rod with and without His-tag also did not sediment on its own, then bound to actin filaments with affinities similar to cofilin without the rod fusion (*Figure 3—figure supplement 3*).

Rabbit skeletal muscle actin and chymotryptic subfragment-1, S1, were purified as described previously (*Spudich and Watt, 1971*; *Margossian and Lowey, 1982*) and stored in liquid nitrogen. Some experiments used G-actin that was further purified by gel filtration column chromatography, yielding identical results.

Before use, an aliquot of frozen stock was thawed for 1 hr on ice and clarified by ultracentrifugation at 80,000 rpm for 5 min at 5°C. Protein concentration was then measured using an Advanced Protein Assay (Cytoskeleton, Denver, CO), using calibrated skeletal actin as the standard.

## High-speed atomic force microscopy

We used a laboratory built high-speed atomic force microscope (HS-AFM) as described previously (*Ando et al., 2013*). HS-AFM imaging was carried out in the tapping mode with small cantilevers (BL-AC10DS-A2, Olympus, Tokyo, Japan) whose spring constant, resonant frequency in water, and quality factor in water were ~0.10 N/m, ~400 kHz, and ~2, respectively. The probe tip was grown on the original tip end of a cantilever through electron beam deposition and was further sharpened using

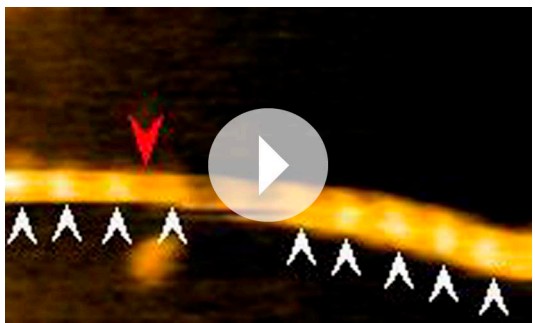

**Video 9.** Severing of actin filaments by cofilin. To show more general view of severing events, in addition to the small number of representative cases shown in **Videos 6–8, 10** different image sequences from different experiments were merged. Sequence numbers are shown in the first 10 frames of each sequence. For color codes of the arrowheads, see the legend to **Video 6**. Conditions: F buffer containing 1 mM ATP (sequences 1–6), 1 mM ADP (sequence 7–8) or 1 mM ATP + 10 mM Pi (sequence 9–10). The concentration of cofilin shown in this video was 75 nM, except in sequences 9 and 10, in which it was 300 and 150 nM, respectively. Z-scale was 0–12 nm. Related to **Figure 6—figure supplement 1**.

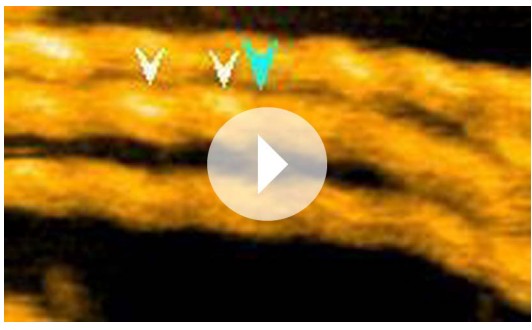

**Video 10.** Severing of actin filaments by cofilin without His-tag. In this video, seven different image sequences from different filaments and experiments were merged. Conditions: F buffer containing 1 mM ATP and 75 nM cofilin, except in sequence 3 in which cofilin concentration was 150 nM. For color codes of the arrowheads, see the legend to **Video 6**. Z-scale was 0–12 nm. Related to **Figure 6—figure supplement 2**.

a radio frequency plasma etcher (PE-2000, South Bay Technology, Redondo Beach, CA) under an argon gas atmosphere (typically at 180 mTorr and 15 W for 3 min). During HS-AFM imaging, the free-oscillation peak-to-peak amplitude of the cantilever ($A_0$) was set to ~2 nm, and the feedback amplitude set point was set at more than $0.9A_0$. Details of the method for HS-AFM imaging are described elsewhere (*Uchihashi et al., 2012*).

## Mica-supported lipid bilayer

We prepared small unilamellar vesicles (SUVs) and mica-supported lipid bilayer as described previously (*Yamamoto et al., 2010*; *Uchihashi et al., 2012*). The typical lipid composition was 1,2-dipamitoyl-*sn*-glycero-3-phosphocholine (DPPC) and 1,2-dipalmitoyl-3-trimethylammonium-propane (DPTAP) at a weight ratio of 9:1. The lipids were purchased from Avanti Polar Lipids (Alabaster, AL). SUVs were dispersed in Milli-Q water at 2 mg/ml and stocked at −20°C. Before use, the SUVs were diluted in 5 mM $MgCl_2$ to 0.5 mg/ml and sonicated with a bath sonicator (AUC-06L, AS ONE, Osaka, Japan) for 1 min. An aliquot of the sonicated SUVs was deposited on the surface of freshly cleaved mica, which had been glued onto a sample stage beforehand, and incubated for more than 3 hr at room temperature (24–26°C) in a humid sealed container to avoid surface drying. Up to 10 sample stages were prepared simultaneously and stored in the sealed container.

## HS-AFM imaging

Before deposition of actin filaments, the surface of the sample stage was rinsed with a large amount of Milli-Q water (~20 µl × five times) to remove excess SUVs and lipid bilayers. Actin filaments were then deposited onto the lipid bilayer using one of the following methods.

G-actin (5–10 µM) was polymerized in F buffer (40 mM KCl, 20 mM PIPES–KOH, pH 6.8, 1 mM $MgCl_2$, 0.5 mM EGTA, 0.5 mM DTT) containing 1 mM ATP for 30 min on ice. The resultant actin filaments were diluted to 0.5–1.0 µM in F buffer containing 1 mM ATP. Water on the lipid bilayer on a sample stage was replaced with 1–2 µl of F buffer containing 1 mM ATP, to which 2 µl of the diluted actin solution was added. After 5–10 min of incubation at room temperature, unattached actin filaments were removed by exchanging the solution with 50 µl of one of the four different F buffer-based observation buffers each containing (i) 1 mM ATP, (ii) 0.1 mM ATP and 1 mM ADP, (iii) 1 mM ADP, 5 U/ml hexokinase, and 10 mM glucose, and (iv) 1 mM ADP and 10 mM Pi. In the case (iii), incubation was continued for 30 min to ensure that all actin subunits within the filaments carried ADP.

Alternatively, G-actin (20 µM) was polymerized in F buffer containing 1 mM ATP and 30 mM $MgCl_2$ for 1 hr at room temperature. After introduction of this solution to the sample stage and 10 min of incubation at room temperature, unattached actin filaments were removed by gently exchanging the solution with F buffer containing 1 mM ATP and 30 mM $MgCl_2$.

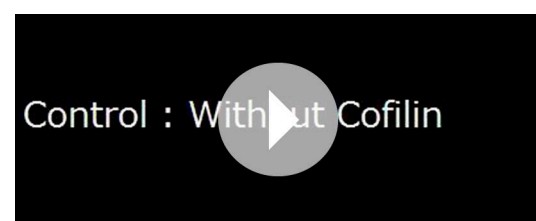

**Video 11.** Severing of actin filaments in the absence or presence of low concentrations of cofilin. Four independent image sequences are merged. Conditions: F buffer containing 1 mM ATP and 0, 10 or 40 nM cofilin without His-tag. In this video, three data sets which represent three cases of the absence or presence of cofilin are sequentially shown and indicated before each sequence begins as (i) Control: Without Cofilin, (ii) 10 nM Cofilin (two different image sequences), and (iii) 40 nM Cofilin. Note that severing of actin filaments was not observed not only in the absence but also in the presence of 10 nM cofilin. In the presence of 40 nM cofilin, severing was infrequently observed. For color codes of the arrowheads, see the legend to *Video 6*. Images were taken at 0.5 frames/s, except in the presence of 40 nM cofilin they were recorded at 0.25 frames/s, and the video is played at 5 frames/s. Z-scale was 0–12 nm. Related to *Figure 6— figure supplement 3*.

**Video 12.** Actin filaments decorated with high concentrations of cofilin. The first three quarter of this video were taken in F buffer containing 1 mM ATP and 1 μM cofilin, and the last one quarter was taken in the presence of 650 nM cofilin. In the presence of 1 μM cofilin, filaments were fully decorated along the length, and no severing was observed. In the presence of 650 nM cofilin, there were some bare zones, and severing occurred near the boundary of the bare zone and the cofilin clusters, regardless of the size of cofilin clusters. For color codes of the arrowheads, see the legend to *Video 6*. Images were taken at 2 frames/s and played at 5 frames/s. Z-scale was 0–12 nm. Related to *Figure 6—figure supplement 4*.

Finally, the sample stage was mounted to the z-scanner of a HS-AFM apparatus and immersed in a liquid cell containing the same observation buffer used in the last step, and HS-AFM imaging was performed.

To follow binding of cofilin to actin filaments, 6 μl of cofilin diluted in the observation buffer was injected into the observation cell during AFM imaging. In some experiments, S1 was added in the observation buffer in the concentration of 20 nM (in cases [i], [ii], and [iii]) or 150 nM (in case [iv]) to identify the polarity of actin filaments.

## Data analyses of HS-AFM image

HS-AFM images were viewed and analyzed using the laboratory built software, Kodec4.4.7.39. In brief, a low-pass filter to remove spike noise and a flattening filter to make the xy-plane flat were applied to individual images. The position and height of the peak within each half helix were determined semi-automatically using the following steps. First, the most probable highest point near a crossover point was selected manually. Second, the actual highest point was automatically determined by searching a 5 × 5 pixel area (typically 7.5 × 7.5 nm$^2$) around the selected point. Third, the peak position was refined based on a center of mass calculation using information on the heights and positions within the 5 × 5 pixel area around the selected point, after which the refined peak position and height were used to represent the peak of the half helix.

The Kodec4.4.7.39 for HS-AFM image viewing and analysis software is coded in Visual C# (Visual Studio 2010, Microsoft, USA) and is available as *Source code 1*. All filters and subroutines for image analysis used in the present study are included in the software. We confirmed the compatibility between the software and computers operated with Windows 7 or 8. Installer of the software, Kodec4_Setup.msi, is available in the subfolder of 'Kodec 4.4.7.39\Setup\Release'. This software should be cited as: Sakashita M, M Imai, N Kodera, D Maruyama, H Watanabe, Y Moriguchi, and T Ando. 2013. Kodec4.4.7.39.

## Electron microscopy

Actin filaments were prepared by polymerization of G-actin (20 μM) in F buffer containing 40 mM KCl, 20 mM PIPES, pH 6.8, 1 mM MgCl$_2$, 0.5 mM EGTA, 0.5 mM DTT, and 1 mM ATP for 30 min on ice. Cofilin or cofilin-rod was added to 1 μM actin filaments in 50 μl of F buffer at a molar ratio of 1:2 or 1:1, and the solutions were mixed by gentle pipetting, after which the mixture was incubated for 3–5 min at room temperature. This mixture was then added to F buffer containing sodium phosphate (pH 6.8) so that the concentration of actin

was 0.5–1 µM and that of Pi was 5 mM, and a drop of this solution was immediately placed on a copper grid. The samples were fixed and negatively stained using a solution containing 1% uranyl acetate and 20 µg/ml bacitracin (*Katayama, 1989*). The fixed samples were dried under an incandescent lamp to form films over the holes of the grid. Electron microscopic data were then acquired using a Hitachi H-7650 transmission electron microscope.

## Acknowledgements

We thank Dr Akira Nagasaki for cooperation in preliminary characterization of actin–cofilin interactions using TIRF microscopy, Dr Shin-ichiro Kojima for help in statistical analysis, and Dr Masafumi Yamada for providing actin. We are also indebted to Drs Keiko Hirose, Makoto Miyata and Taro Noguchi for comments on the manuscript. This work was supported in part by Grants-in-aid from the Ministry of Education, Culture, Sports, Science and Technology for TU, a PRESTO grant from the Japan Science and Technology Agency (JST) to NK, and a CREST grant from JST to TA.

## Additional information

### Funding

| Funder | Author |
| --- | --- |
| Ministry of Education, Culture, Sports, Science, and Technology | Taro QP Uyeda |
| Japan Science and Technology Agency | Noriyuki Kodera, Toshio Ando |

The funders had no role in study design, data collection and interpretation, or the decision to submit the work for publication.

### Author contributions

KXN, Acquisition of data, Analysis and interpretation of data, Drafting or revising the article; NK, Conception and design, Acquisition of data, Analysis and interpretation of data, Drafting or revising the article; EK, Acquisition of data, Analysis and interpretation of data; TA, TQPU, Conception and design, Drafting or revising the article

## Additional files

### Supplementary file

• Source code 1. The Kodec4.4.7.39 for HS-AFM image viewing and analysis software is coded in Visual C# (Visual Studio 2010, Microsoft, USA).

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
