## [Decision Letter]

Thank you for sending your work entitled “Cofilin-induced unidirectional cooperative conformational changes in actin filaments revealed by high-speed AFM” for consideration at *eLife*. Your article has been favorably evaluated by John Kuriyan (Senior editor), a guest Reviewing editor, and 3 reviewers.

The following individuals responsible for the peer review of your submission have agreed to reveal their identity: Brad Nolen (Reviewing editor); Roberto Dominguez and Thomas Pollard (peer reviewers). A further reviewer remains anonymous.

The Reviewing editor and the reviewers discussed their comments before we reached this decision, and the Reviewing editor has assembled the following comments to help you prepare a revised submission.

Here, Ngo et al., applied high-speed AFM to visualize interactions of cofilin with actin filaments. This method offers unprecedented resolution for studying the process in real time, so there is no question about the technical advance over previous methods. They show that cofilin forms clusters along actin filaments and discovered that the shortened helical pitch propagates to the neighboring 'cofilin-free' zone towards the filament pointed end. Furthermore, they demonstrate that cofilin clusters grew unidirectionally towards the filament pointed end. Finally, they show that severing most frequently occurs within the cofilin clusters (>50%), while 1/3 of severing events were observed to occur near these clusters. Because of the critical importance of cofilin severing in nearly all actin cytoskeleton activities, these findings have general relevance and interest for the readers of *eLife*. However, additional experiments are required to support the authors' conclusions:

1) Control experiments are needed to validate the cofilin constructs used. Specifically, the data in Figure 3—figure supplement 2, should be replotted as binding density versus free cofilin, and the figure should include binding isotherms for wild type cofilin, both with His-tag cleaved and with His tag uncleaved. The authors should indicate the molar concentrations of components used instead of the ratios in Figure 3—figure supplement 1 and show a control with cofilin-rod alone at the maximum concentration used in the assay to show that this protein is not sedimenting on its own. The authors also should clarify why at 1:1, there is more actin in the supernatant than in the pellet, and then at 2:1 most of the actin goes back to being in the pellet.

2) The authors should repeat experiments done with a mix of Pi and ATP with only Pi. In Figures 4 and 5, adding Pi is reasonable, because it will result in ADP-Pi -bound subunits. However, there is no reason to add ATP because the bound nucleotide does not exchange on polymerized actin, and addition of ATP makes it impossible to determine the orientation of filaments since ATP dissociates S1. Therefore, the authors should repeat this experiment with Pi and no ATP, so that S1 decoration and filament orientation can be determined. In addition, the authors should clarify the issue of the nucleotide state of the filament throughout the manuscript, including in setting up the rationale for the experiments and in the interpretation of the results. For example, in the Discussion section, the authors state that “This tendency of cofilin clusters to grow unidirectionally was unaffected when most of the filament subunits carried ADP or either ATP or ADP+Pi”, but the filaments would not have bound ATP under any of the conditions that they tested.

3) The authors should explicitly specify in the figure legends the concentrations of cofilin used in the severing assay. If the concentrations are not at or near the optimal severing concentration for cofilin (Andrianantoandro, 2006), the severing experiments should be repeated at the optimal concentration. In addition, severing data should also be more precisely analyzed to indicate where severing occurs. The new analysis should allow the authors to make statements about where within the cluster severing is occurring (e.g. closer to the ends or evenly distributed throughout the clusters). The authors should also clarify how many severing events were analyzed (i.e. does the 'frequency' correspond to absolute numbers of the severing events?). Disagreement with previous severing models/results should be more clearly addressed.

The authors should also address the following issues in the revised manuscript:

1) The strategy for the alpha-actinin-cofilin fusion construct should be better explained. Please be more specific about the region of alpha-actinin that was fused (what is meant by “half of an a-actinin rod”?) State what amino acid numbers were used. Cite literature that supports the argument that the construct used will form a 6 angstrom long rod. Also, please address the possibility that the fusion protein could possibly form antiparallel dimers via spectrin repeat interactions.

2) The observation that the conformational change in the filament spreads asymmetrically is an important result but is unexpected based on thermodynamic considerations. The authors are encouraged to repeat the analysis of this data using a blinded observer.

3) The extent of the propagation of the supertwisted conformation from a cofilin cluster to the neighboring bare zone at the pointed end should be precisely described. How many repeats become supertwisted after a cofilin cluster? Also, is this a structural effect that decreases gradually or abruptly? The authors are encouraged to quantitatively address these questions.

---

## [Author Response]

*1) Control experiments are needed to validate the cofilin constructs used. Specifically, the data in*
Figure 3—figure supplement 2*, should be replotted as binding density versus free cofilin, and the figure should include binding isotherms for wild type cofilin, both with His-tag cleaved and with His tag uncleaved. The authors should indicate the molar concentrations of components used instead of the ratios in*
Figure 3—figure supplement 1
*and show a control with cofilin-rod alone at the maximum concentration used in the assay to show that this protein is not sedimenting on its own. The authors also should clarify why at 1:1, there is more actin in the supernatant than in the pellet, and then at 2:1 most of the actin goes back to being in the pellet*.

According to this suggestion, we repeated cosedimentation experiments (Figure 3—figure supplement 1) and replotted the data on cofilin with and without His tag as a function of free cofilin concentration (Figure 3—figure supplement 2). It is evident that both cofilin with and without His tag and cofilin-rod with and without His tag do not sediment on its own (Figure 3—figure supplement 3). Cofilin with and without His tag appeared to have slightly different affinities for actin (Figure 3—figure supplement 2), but we did not pursue that issue further, since that is not a main topic of this paper.

In the first version, the 1:1 data point was strange, which was because we loaded that pair of supernatant and pellet in a wrong order. We apologize for not mentioning that in the original manuscript, which confused the reviewers.

*2) The authors should repeat experiments done with a mix of Pi and ATP with only Pi. In*
Figures 4 and 5*, adding Pi is reasonable, because it will result in ADP-Pi -bound subunits. However, there is no reason to add ATP because the bound nucleotide does not exchange on polymerized actin, and addition of ATP makes it impossible to determine the orientation of filaments since ATP dissociates S1. Therefore, the authors should repeat this experiment with Pi and no ATP, so that S1 decoration and filament orientation can be determined. In addition, the authors should clarify the issue of the nucleotide state of the filament throughout the manuscript, including in setting up the rationale for the experiments and in the interpretation of the results. For example, in the Discussion section, the authors state that “This tendency of cofilin clusters to grow unidirectionally was unaffected when most of the filament subunits carried ADP or either ATP or ADP+Pi”, but the filaments would not have bound ATP under any of the conditions that they tested*.

Based on this constructive comment, we observed growth of cofilin clusters in the presence of ADP and 10 mM Pi, but no ATP. The results were clear, enabling us to conclude that the clusters grew unidirectionally to the P-ends regardless of the nucleotide states of actin subunits. We have thus replaced the old Figure 4 with this new set of data.

In addition, we have re-examined the nucleotide states of actin subunits during the experiments, and made revisions accordingly.

*3) The authors should explicitly specify in the figure legends the concentrations of cofilin used in the severing assay. If the concentrations are not at or near the optimal severing concentration for cofilin (Andrianantoandro, 2006), the severing experiments should be repeated at the optimal concentration. In addition, severing data should also be more precisely analyzed to indicate where severing occurs. The new analysis should allow the authors to make statements about where within the cluster severing is occurring (e.g. closer to the ends or evenly distributed throughout the clusters). The authors should also clarify how many severing events were analyzed (i.e. does the 'frequency' correspond to absolute numbers of the severing events?). Disagreement with previous severing models/results should be more clearly addressed*.

We apologize for failing to specify the concentrations of cofilin used in the severing assays. It was 75 nM, and it is now written in the legend to Figure 6. 75 nM is significantly higher than that reported to be optimum by [2] (10 nM), and therefore, we observed severing in the presence of lower concentrations, down to 10 nM. However, we did not detect severing in the presence of 10 cofilin, and only very infrequently in 20 nM. In the presence of cofilin higher than 500 nM, binding was so rapid that it was difficult to follow the growth of cofilin clusters even by HS-AFM, but we did confirm that severing is very infrequent in the presence of 500 nM cofilin and almost not at all in 1 µM. Thus, we also found that intermediate concentrations of cofilin (around 40-200 nM) are optimum for severing, although we do not have quantitative data. This is qualitatively consistent with Andrianantoandro and Pollard’s finding. However, our optimum concentrations are higher by several-fold, and we can only speculate that this discrepancy may be due to differences in experimental conditions. In Andrianantoandro’s experiments, actin filaments were immobilized by small number of NEM-myosin molecules, while in our experiments, actin filaments were attracted to the positively charged lipid membrane. Furthermore, in Andrianantoandro’s experiments, actin filaments were chemically labeled with fluorophores. All these may affect the state of actin filaments in different manners. In addition, movements of actin filaments are more or less restricted to two-dimension on the lipid surface under our conditions. Furthermore, Andrianantoandro reported that the severing activity is dependent on filament length, and it is plausible that our filaments were shorter than Andrianantoandro’s.

We have also performed a more careful analysis of severing position with respect to the cofilin clusters (in the Discussion section), which supports our original conclusion that severing occurs primarily near the boundary of cofilin clusters and bare zones, but not deeper inside the clusters or in bare zones far from clusters.

Regarding the number of severing events, the numbers shown in Figure 6 are the number of events. To make this point clear, we relabeled the x-axis as “Number of events” instead of “Frequency”.

At 40 nM cofilin, we did observe infrequent severing (N=22). In the majority of the cases, we detected a short cofilin cluster near the severing site. This finding strongly supports our conclusion that cofilin clusters, but not singly or sparsely bound cofilin molecules, accelerate severing of the filaments. This point is now highlighted in the Discussion section, referring to contradictory earlier reports including that by [2].

*The authors should also address the following issues in the revised manuscript*:

*1) The strategy for the alpha-actinin-cofilin fusion construct should be better explained. Please be more specific about the region of alpha-actinin that was fused (what is meant by “half of an a-actinin rod”?) State what amino acid numbers were used. Cite literature that supports the argument that the construct used will form a 6 angstrom long rod. Also, please address the possibility that the fusion protein could possibly form antiparallel dimers via spectrin repeat interactions*.

We have added a more detailed description on the actinin-rod to the Materials and methods section, including references for the monomeric nature and the length of the rod. The references are also cited in the main text. In addition, we wrote erroneously that the rod is ∼6 nm long in the original manuscript.

*2) The observation that the conformational change in the filament spreads asymmetrically is an important result but is unexpected based on thermodynamic considerations. The authors are encouraged to repeat the analysis of this data using a blinded observer*.

We had two researchers not involved in this study judging the polarity of the filament based on the appearance of transiently bound S1 molecules. In all of the cases tested, their judgment agreed with the original judgment.

*3) The extent of the propagation of the supertwisted conformation from a cofilin cluster to the neighboring bare zone at the pointed end should be precisely described. How many repeats become supertwisted after a cofilin cluster? Also, is this a structural effect that decreases gradually or abruptly? The authors are encouraged to quantitatively address these questions*.

We have measured half helical pitches of the neighbors second from the cofilin clusters, and found that they were nearly normal (Table 1). Unfortunately, our current measurement algorithm only allows for the measurement of half helical pitches between peaks. It is also impossible to determine where exactly the cofilin cluster ends. For those two reasons, it is difficult to obtain more detailed information about how far the conformational change propagates into neighboring bare zones and whether the structural effect decreases abruptly or gradually. However, the fact that the second neighboring bare zone had almost the normal pitch length seems to suggest that the length of supertwisted neighboring bare zone is shorter than one half helix. This speculation is added in the revised manuscript.